# Design of a Compact Circularly Polarized Implantable Antenna for Capsule Endoscopy Systems

**DOI:** 10.3390/s24123960

**Published:** 2024-06-19

**Authors:** Zhiwei Song, Xiaoming Xu, Youwei Shi, Lu Wang

**Affiliations:** 1The School of Electrical Engineering, Xinjiang University, Huarui Street 777#, Shuimogou District, Urumqi 830047, China; xxm16@stu.xju.edu.cn (X.X.); 107552103985@stu.xju.edu.cn (Y.S.); 2The 58th Research Institute of China Electronics Technology Group Corporation, Wuxi 214072, China; wanglu_william@163.com

**Keywords:** antenna, biomedical telemetry, circular polarization, wireless capsule endoscopy system

## Abstract

This research proposes a miniature circular polarization antenna used in a wireless capsule endoscopy system at 2.45 GHz for industrial, scientific, and medical bands. We propose a method of cutting a chamfer rectangular slot on a circular radiation patch and introducing a curved radiation structure into the centerline position of the chamfer rectangular slot, while a short-circuit probe is added to achieve miniaturization. Therefore, we significantly reduced the size of the antenna and made it exhibit circularly polarized radiation characteristics. A cross-slot is cut in the GND to enable the antenna to better cover the operating band while being able to meet the complex human environment. The effective axis ratio bandwidth is 120 MHz (2.38–2.50 GHz). Its size is π × 0.032λ_0_^2^ × 0.007λ_0_ (where λ_0_ is the free-space wavelength of at 2.4 GHz). In addition, the effect of different organs such as muscle, stomach, small intestine, and big intestine on the antenna when it was embedded into the wireless capsule endoscopy (WCE) system was further discussed, and the results proved that the WCE system has better robustness in different organs. The antenna’s specific absorption rate can follow the IEEE Standard Safety Guidelines (IEEE C95.1-1999). A prototype is fabricated and measured. The experimental results are consistent with the simulation results.

## 1. Introduction

Wireless implantable medical equipment is broadly applied in medical diagnostics and healthcare. These medical devices along with other biotechnologies such as wireless endoscopic systems can send information about the diagnosis of a patient to a medical center that can make treatment decisions. For example, wireless endoscopic systems can visualize the intestinal pathway better than conventional endoscopic systems [1]. As a wireless medical device specializing in the diagnosis of human gastrointestinal disorders, wireless capsule endoscopes usually consist of several components: an implantable antenna, a battery pack, a transceiver, an LED light, and a camera [2]. Research has shown that after swallowing a capsule, it usually remains in the body for around 10 h [3]. Thus, the electricity provided by batteries is particularly important. It is well known that the power supplied by the battery is proportional to its size, requiring that the design of other components must be as small as possible. Therefore, the miniaturization of implantable antennas is essential in capsule endoscopy systems.

Antennas for endoscopic systems are a more challenging task compared to implanted antennas with fixed implantation locations [4,5]. It is well known that the human digestive tract consists of organs like the large intestine, colon, small intestine, and stomach, and the dielectric properties vary from organ to organ. To obviate the bandwidth shift and improve the ability of capsule endoscopy to diagnose diseased tissues or organs, the capsule antenna passing through the human GI tract must have stable characteristics for the continuity and accuracy of communication.

The majority of implantable antennas tend to be linearly polarized antennas, which are susceptible to multi-path disturbances and mismatch of polarization due to the random orientation and location of endoscopic systems in the gastrointestinal tract. Circularly polarized (CP) antennas are an essential requirement for high-quality communication with external devices for reduced multipath distortion and increased BER. In 2019, researchers proposed an ultra-wideband ring antenna that is conformal to the capsule’s outer wall, which saves more space for the battery and the associated circuits [6]. In this work, the authors built a human model close to the real size of the body’s alimentary canal portion and placed the capsule antenna at its center for simulation. The antenna operates in the Industrial, Scientific, and Medical (ISM) band with omnidirectional radiation characteristics. In [7,8,9,10,11], the authors designed antennas that co-adhesive with the external shell of the capsule. In [9], a meander symmetric fabric patch, folding technique, and sagittal current inversion technique were used to develop a novel 915 MHz ring antenna, which coincides with the inside shell of the capsule. Since such conformal antennas are usually made of flexible materials and fit snugly into the capsule shell, they are susceptible to interference from gastric juices and other components of the capsule (e.g., the bio-coating and the capsule), affecting the original permittivity and thus altering the antenna performance. Therefore, embedded structures are another mainstream design direction, in which the antenna is positioned in the capsule chamber with other equipment, and the planar dimensions on the antenna are a little smaller than the inner diameter of the capsule. This layout requires the antenna to be very compact so that it can be easily mounted into the capsule [12]. Embedded antennas can be designed as classical spiral structures [13] and patch-shaped antennas [14]. In [15,16,17], the authors designed helical structures to minimize the antenna dimensions. In [15], the authors proposed a multilayer spiral antenna that can be used in the ISM band for capsule endoscopy systems. In [18,19,20,21,22], the authors printed the antenna directly on a dielectric substrate not only to obtain a smaller dimension but also to allow a more flexible adjustment of the antenna’s shape. In [19], the authors cut slots in both the radiating patch and ground plane (GND) to minimize the dimensions of the antenna and increase its impedance bandwidth, thus facilitating the embedding of the antenna in the capsule system.

This paper proposes a miniature circularly polarized implantable antenna for capsule endoscopy systems. The antenna consists of a radiation patch, a GND, a circular dielectric substrate, and a superstrate. The substrate and superstrate are made of Rogers RO 6010 (*ε_r_* = 10.2, tan*δ* = 0.001). We propose a method of cutting a chamfer rectangular slot on a circular radiation patch and introducing a curved radiation structure into the centerline position of the chamfer rectangular slot, while a short-circuit probe is added to achieve miniaturization. Therefore, we significantly reduced the size of the antenna and made it exhibit circularly polarized radiation characteristics. A cross-slot is cut in GND to enable the antenna to better cover the operating band while being able to meet the complex human environment. Its size is π × 0.032λ_0_^2^ × 0.007λ_0_ (44.7 mm^3^). The antenna is installed in a capsule endoscope model, and the simulation is carried out in a multilayer human tissue. The designed antenna is fabricated and tested. The results indicate that the antenna possesses simple geometry and good radiation characteristics. The actual effective AR bandwidth is 120 MHz (2.38 to 2.50 GHz, 5%). In addition, the effect of different organs such as muscle, stomach, small intestine, and big intestine on the antenna when it was embedded into the wireless capsule endoscopy (WCE) system was further discussed, and the results proved that the WCE system has better robustness in different organs, with its use following the IEEE Standard Safety Guidelines (IEEE C95.1-1999) [23].

## 2. Antenna Design and Optimization

### 2.1. Antenna Geometry

To achieve a compact antenna compatible with wireless capsule endoscopy system applications, slotting techniques, a selection of higher dielectric constant materials, and short-circuited probes are used. Based on the techniques, the scheme of the proposed antenna is illustrated in Figure 1. The radiation patch is a circle with a radius of 4 mm. A square with triangular cuts in the upper left and lower right corner is cut in the radiation patch with the top center as the center of the circle. A meander structure in the form of an S-shape is incorporated on the left side, and a winding structure is incorporated on the right side. The meandering structure in the radiation patch increases the length of the current path and effectively reduces the dimensions of the antenna. Table 1 lists the exact dimensions of the antenna parameters and the antenna with a size of 44.7 mm^3^ (π × 4^2^ × 0.889 mm^3^). The CP characteristics can be obtained by adding a short-circuit probe and adjusting the feed position. The effective AR band can be improved by adjusting the slot in the GND. For additional bandwidth improvement, the thickness of the dielectric substrate is appropriately adjusted. The feeds with a radius of 0.3 mm are located at the left side of the GND to excite the antenna and are adjusted to achieve good impedance matching and circular polarization behavior. The patch and GND are located at the top and bottom surfaces of the dielectric substrate, respectively, and they played a major role in reducing the size of the antenna. The housing for the capsule endoscopy system is made of the bio-compatible material polyimide, which is used to protect the system components from direct contact with the human body.

The antenna system is implanted into a multi-layer simulation of human tissue for simulation and analysis. The antenna is embedded in the small intestine human model with a radius of 20 mm and a height of 60 mm. To better simulate the working environment of the system in the human body, the antenna is located on the right side along the median axis, as illustrated in Figure 1d. This is because the position of the human organ, the small intestine, is more obviously biased towards the right side of the human body. The outer layer of the small intestine model is a muscle model with a radius of 52 mm and a thickness of 32 mm, the outer layer of the muscle model is a fat model with a radius of 56 mm and a thickness of 4 mm, and the outermost layer is a skin model with a radius of 60 mm and a thickness of 4 mm. The electrical properties of the simulated environment are the same as those of human tissues, and Table 2 presents the dielectric constants and conductivities of these human tissues at a 2.45 GHz frequency, e.g., the skin tissue at a 2.4 GHz frequency has a dielectric constant of 41.3 and a conductivity of 1.464 S/m [24]. We implanted the system into different human digestive systems for simulation testing. The simulation environment is shown in Figure 1d, with the difference being that the electrical characteristic parameters of the small intestine are replaced by those of other tissues. The simulation results are illustrated in Figure 2, in which the S_11_ plot shows the robustness of the system, which can well cover the 2.45 GHz frequency band for different human organs. The axial ratio (AR) plot can be observed that the antenna has circular polarization characteristics in different human organs, so the design has high stability and can meet the complex working environment. The antenna system is set on the right side of the central axis of the multilayer human tissue model with an implantation depth of 30 mm.

### 2.2. Simulation Settings and Environments

As shown in Table 3, four antenna optimization schemes are designed.

Step 1: only the circular radiation patch without slots, but a short-circuit probe is added at the upper right.

Step 2: A square with triangular cut-outs at the upper left and lower right corners is slotted in the center of the circular patch, and a square waveform winding structure is added on the right side. The addition of chamfered rectangular slots is to change the current path of the radiation patch and enable the antenna to achieve circularly polarized radiation characteristics. The addition of a meandering structure is aimed at miniaturizing the antenna and providing additional degrees of freedom for adjusting the center frequency of the antenna.

Step 3: Based on the second step, we formed an S-shaped winding structure in the left area of the blank part of the radiation patch. The addition of an S-shaped winding structure is aimed at miniaturizing the antenna and providing additional degrees of freedom for adjusting the center frequency of the antenna.

Step 4: we proposed cutting a cross-slot in the GND to adjust the AR center frequency.

The simulated S_11_ of the above four cases is shown in Figure 3, and the following conclusions are obtained.

(1) The addition of a shorting probe introduces additional inductance and capacitance, and changes the current distribution, resulting in a significant reduction in antenna size.

(2) The meandering structure further reduces antenna resonance frequency.

(3) Slot-cutting techniques are used to increase the working length of the antenna and to shift the operating frequency to lower bands.

(4) Slotting on GND affects the current path and distribution of the antenna, which will change the radiation characteristics, such as CP.

### 2.3. Simulation and Analysis of Key Parameters

In this section, the parameters of the designed antenna are investigated, and the effect of some parameters on the impedance matching and circular polarization of the antenna is described. In order to shorten the testing time, this section performs simulation tests only in the small intestine model, and the modelling results are shown in Figure 4, Figure 5 and Figure 6. The following conclusions can be drawn.

(1) The thicknesses of the substrate and superstrate change the inductance and capacitance of the antenna structure, and the thickness of H1 and H2 affects the impedance bandwidth to some extent, with a lesser effect on the AR frequency, as shown in Figure 4.

(2) Uniform incremental changes in all slot widths and lengths result in equal variations in the equivalent slot capacitance. So, roughly equal offsets are observed over the impedance bandwidth and in the AR band, as shown in Figure 5 and Figure 6.

(3) By cutting cross slots of specific shapes and sizes on the GND, it is possible to influence the operating frequency of the antenna and adjust the antenna’s radiation patterns and polarization direction.

In addition, it is considered that after the capsule system enters the human body, the system undergoes an uncontrollable change in the bit position. Therefore, considering these factors, as Figure 7 shows the impedance matching S_11_ and axial ratio plots at different postures, it can be clearly seen that the impedance bandwidth is almost unchanged at different postures, with large differences only in amplitude. Moreover, the axial ratio bandwidth undergoes varying degrees of changes but still stably covers the operating frequency band. Therefore, the designed capsule endoscopy system maintains excellent stability in various positions.

## 3. Analysis of Simulation Results

### 3.1. Research on Antenna Working Mechanism

The antenna designed in this paper is modified from a microstrip antenna. We choose a circular radiation patch first, with the slotting method and the shorting post being used to extend the effective current path, decrease the dimensions of the antenna, and introduce a 90-degree phase difference across the two orthogonal components of the antenna’s radiation. Therefore, a miniaturized antenna with CP characteristics is gained.

To understand more about the operating principle of the antenna, Figure 8 displays the surface current profiles of the antenna in different phases at 2.4 GHz. At 0° and 180°, most of the surface currents are concentrated on both sides of the slot, but the direction of the currents in each phase is opposite. At 90° and 270°, most of the surface currents are concentrated on the top side of the patch and GND, and the directions are opposite. The cumulative effect of these different phases of surface currents produces circular polarization [25].

### 3.2. Research on Antenna Radiation Characteristics

The 3D radiation pattern of the antenna is simulated in a multilayer body model (the center is the small intestine tissue), as illustrated in Figure 9a. The simulated AR vs. Theta is shown in Figure 9b. It is clear that from when Theta is from −12 degrees to 84 degrees, the antenna has good CP characteristics. The simulated LHCP and RHCP are shown in Figure 9c. The far-field gain at the 2.45 GHz frequency point of the antenna is −26.7 dBi, which maintains a good far-field gain in the test environment and meets the operating requirements of the implantable medical monitoring system.

The simulated specific absorption rate (SAR) distribution of the implantable antenna designed in this paper is shown in Figure 10. Assuming that the antenna’s transmitting power is 1 W, the maximum average SAR value of 1 g at 2.4 GHz is 216 W/kg. Therefore, the maximum transmitting power must be less than 7.4 mW, which complies with the IEEE C95.1-1999 standard. The designed antenna was simulation-tested and its SAR value fully complies with the international safety standards, which ensures that the electromagnetic radiation to the human body remains within the safe range under various conditions of use [23].

## 4. Analysis of Measurement Results

The physical diagram of the antenna is shown in Figure 11a. Physical measurements of the antenna are carried out by using the Agilent Vector Network Analyzer (PNA-X) and the microwave darkroom developed by Electronics Forty One (EFXI), as illustrated in Figure 11b.

The bowl used in the test was 30 cm in diameter and 15 cm in height. The capsule was implanted at a depth of 3 cm, consistent with the simulation setup, in the center of the bowl’s cross-section. In addition, we compared the simulation results of the capsule in the small intestine, large intestine, and stomach environments to demonstrate that the antenna can work properly in the entire digestive system.

The simulated and tested S_11_ is shown in Figure 12. This test was performed only in fresh minced pork tissue, where the bandwidth was 0.67 GHz (2.02–2.69 GHz) and the AR bandwidth was 0.28 GHz (2.26–2.54 GHz). Figure 13 compares the simulated and measured E-plane (Phi = 0 degree) radiation patterns, where the simulated and measured realized gain peaks are −26.7 dBi and −27.8 dBi, respectively. The simulated and measured ones have more significant errors.

In general, errors are always present, and one of the reasons may be that the measurement environment is close but not exactly the same as the simulated environment, lacking the outer layer organization. Another reason is that errors introduced during the manufacturing process, like the injection depth and air gap, separate the antenna from the cover layer.

Table 4 lists the references and the main performance metrics of the antenna designed in this paper to visualize its performance. In summary, the antenna in this paper has a small size and good radiation characteristics.

## 5. Conclusions

In this paper, we designed a compact implantable antenna for capsule endoscopy systems. We analyzed the working principle of the antenna in detail and obtained a set of more desirable antenna parameters. In order to verify the actual radiation performance of the antenna, we fabricated a prototype and performed measurements in minced pork. The measurement results showed that the −10 dB bandwidth is fully compatible with the 2.45 GHz band and that the radiation direction map has some symmetry. The antenna’s size is only 44.7 mm^3^, and its good radiation characteristics make it ideal for capsule endoscopy systems. In addition, we believe that the next step to improve the antenna design could be to adopt broadband technology, super materials, and miniaturization techniques to increase bandwidth and reduce size. Meanwhile, advanced simulation tools and optimization algorithms can be used for continuous experimental validations and iterative improvements.

## Figures and Tables

**Figure 1 sensors-24-03960-f001:**
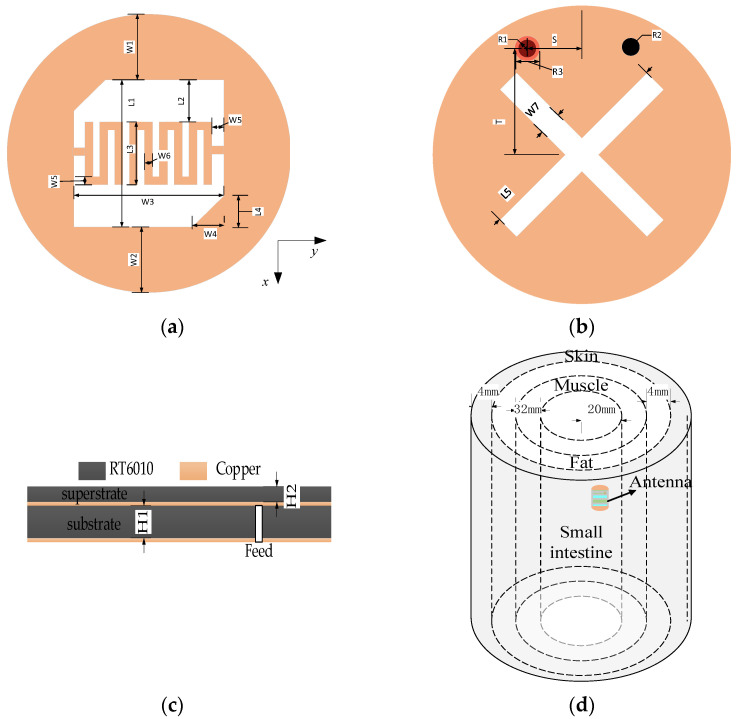
The proposed antenna (not to scale). (**a**) Patch with slots. (**b**) GND with two crossed rectangles. (**c**) Lateral view with symmetrical coaxial feed port. (**d**) Simulator environment.

**Figure 2 sensors-24-03960-f002:**
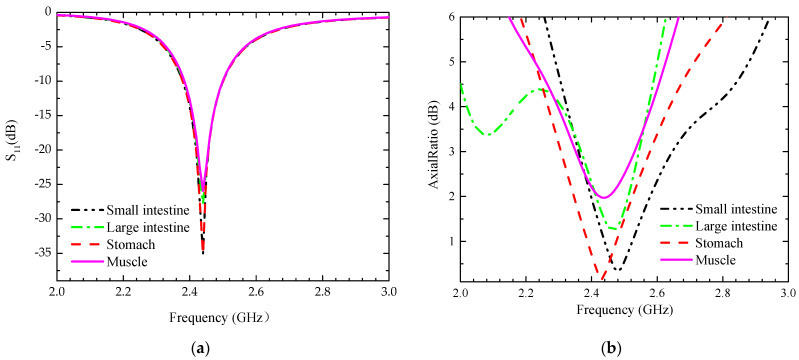
Simulation results at different frequencies. (**a**) S_11_, (**b**) AR.

**Figure 3 sensors-24-03960-f003:**
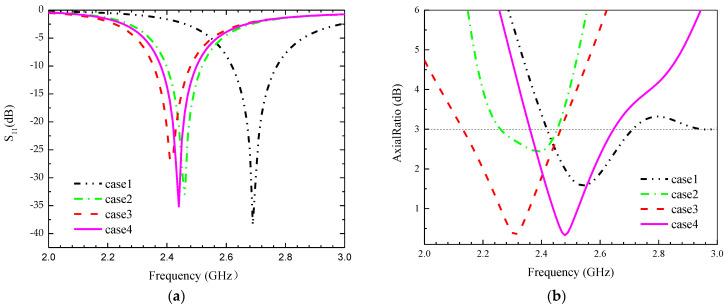
Simulation results for the four cases in S_11_ and AR. (**a**) S_11_, (**b**) AR.

**Figure 4 sensors-24-03960-f004:**
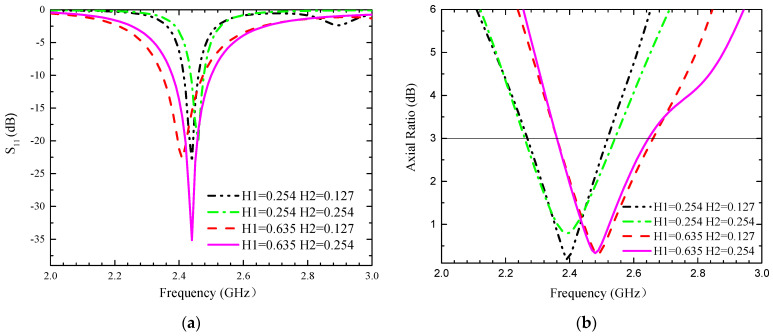
Influence of different parameters H1 and H2 on the proposed antennas. (**a**) S_11_, (**b**) AR.

**Figure 5 sensors-24-03960-f005:**
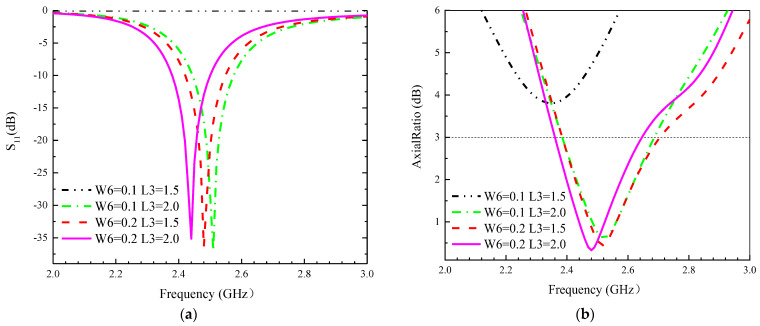
Influence of different parameters W6 and L3 on the proposed antennas. (**a**) S_11_, (**b**) AR.

**Figure 6 sensors-24-03960-f006:**
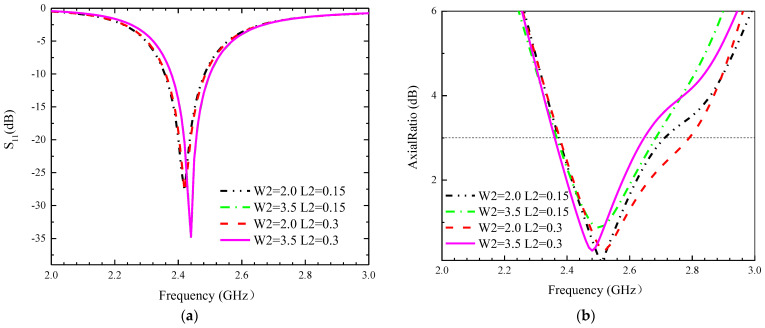
Influence of different parameters W2 and L2 on the proposed antennas. (**a**) S_11_, (**b**) AR.

**Figure 7 sensors-24-03960-f007:**
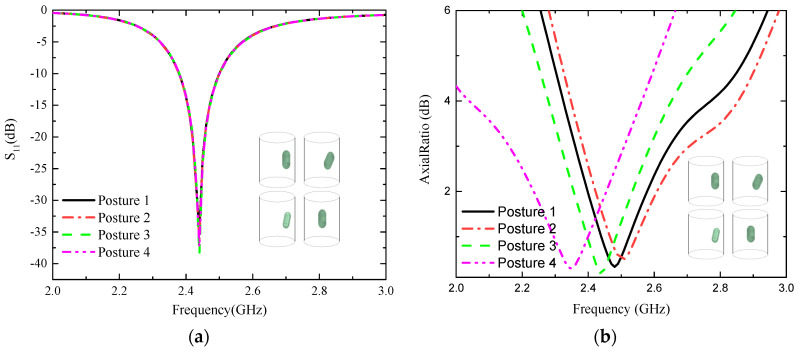
Effect of different bit positions on the proposed antenna. (**a**) S_11_, (**b**) AR.

**Figure 8 sensors-24-03960-f008:**
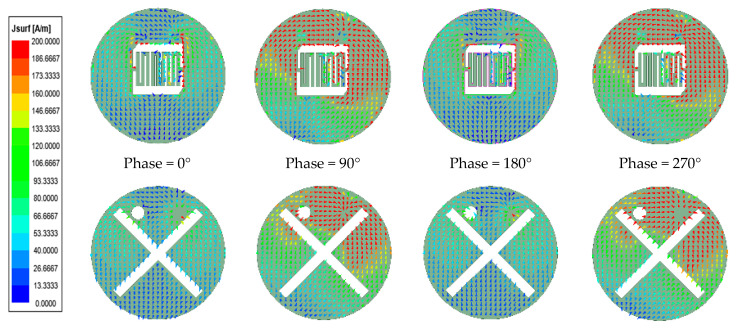
Current distribution on antenna’s surface.

**Figure 9 sensors-24-03960-f009:**
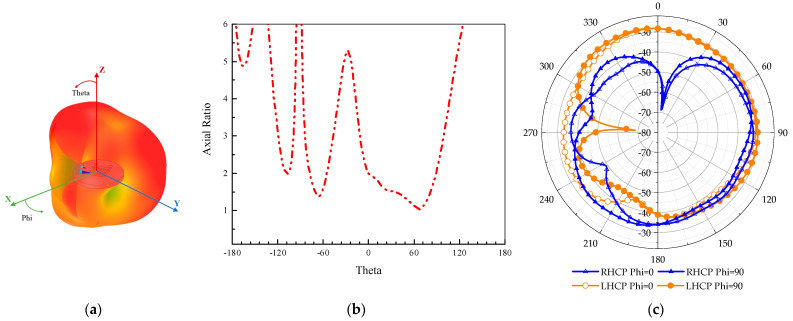
The simulated 3D radiation pattern, AR, and the LHCP and RHCP at 2.4 GHz. (**a**) Three-dimensional radiation pattern. (**b**) AR. (**c**) LHCP and RHCP.

**Figure 10 sensors-24-03960-f010:**
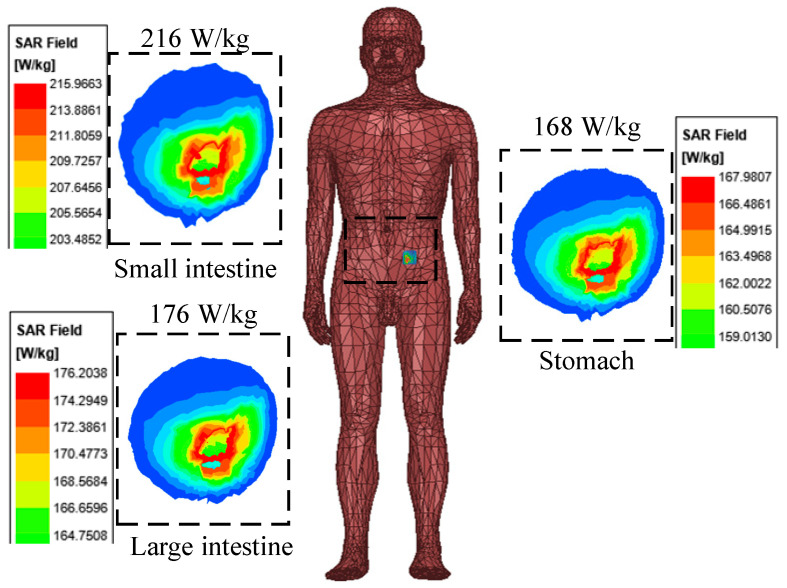
The simulated SAR distribution in skin model (1 g) at 2.45 GHz. The simulation results in the dashed box show the maximum specific absorption rates of the antenna in the small intestine, large intestine, and stomach, which are 216 W/Kg, 176 W/Kg, and 168 W/Kg, respectively.

**Figure 11 sensors-24-03960-f011:**
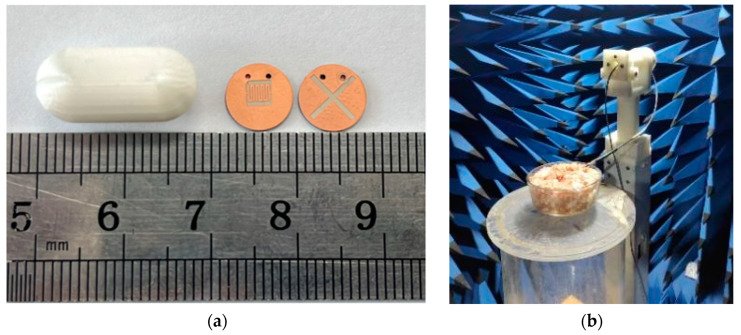
Physical view of the antenna and photographs of the antennae when it is tested in minced pork. (**a**) Antenna and capsule; (**b**) experimental glimpse.

**Figure 12 sensors-24-03960-f012:**
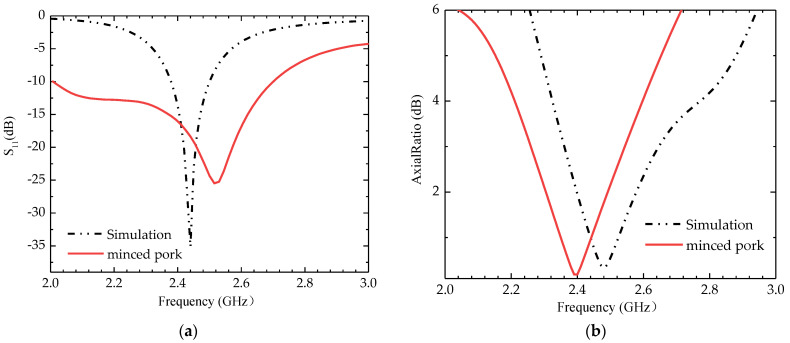
Comparison of simulated and measured S_11_ and AR under different conditions. (**a**) S_11_, (**b**) AR.

**Figure 13 sensors-24-03960-f013:**
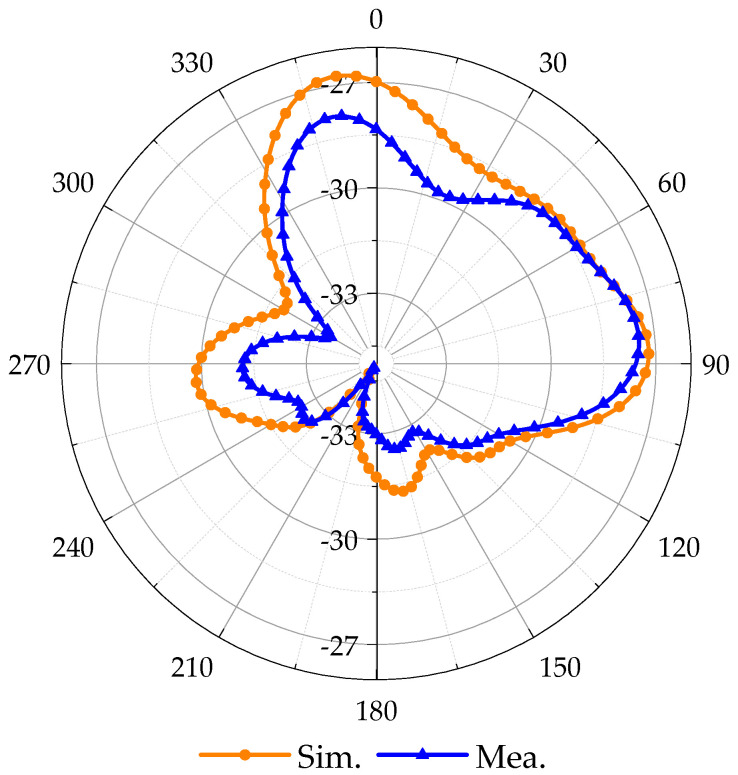
Comparison of simulated and measured E-plane (Phi = 0°) radiation patterns.

**Table 1 sensors-24-03960-t001:** Geometric parameters optimized for the proposed antenna (all the dimensions are in mm).

Parameter	Value	Parameter	Value	Parameter	Value
*W*1	2.1	*W*2	2.9	W3	3
*W*4	0.3	*W*5	0.2	*W*6	0.2
*W*7	0.6	*L*1	3	*L*2	0.5
*L*3	2	*L*4	0.4	*L*5	7.0
*R*1	0.3	*R*2	0.3	*R*3	0.4
*S*	1.2	*T*	2.4	H1	0.635
H2	0.254				

**Table 2 sensors-24-03960-t002:** Dielectric constant and conductivity of different organs in the 2.45 GHz band.

Tissue/Frequency	2.45 GHz	
	*ε_r_*	σ (S/m)
Skin	41.3	1.464
Fat	5.28	0.104
Muscle	52.729	1.73
Small Intestine	54.43	3.17
Large Intestine	53.88	2.04
Stomach	62.16	2.21

**Table 3 sensors-24-03960-t003:** Four scenarios of radiation surface and ground plane in antenna design process.

Case	(1)	(2)	(3)	(4)
Top View	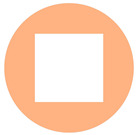	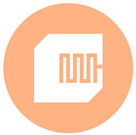	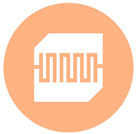	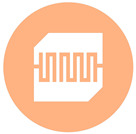
Bottom View	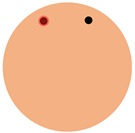	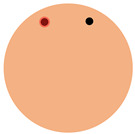	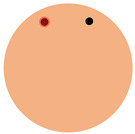	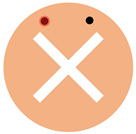

**Table 4 sensors-24-03960-t004:** Proposed antenna in comparison with recent works.

Ref.	Capsule Size (mm)	Freq. (GHz)	Bandwidth (MHz)	Gain (dBi)	SAR (1-g)	CP
[6]	27 × 11	0.433	795	−35	225.4	No
[8]	24 × 11	0.433	260	-	-	No
[10]	26 × 11	2.45	320	−29.1	368.7	Yes
[15]	26 × 11	2.45	960	-	180.7	Yes
[18]	12 × 5	2.45	50	−9.7	596.3	No
[19]	11 × 10	2.45	520	−26.4	712.1	No
This Work	17 × 9	2.45	120	−26.7	216	Yes

## Data Availability

Data are unavailable due to privacy or ethical restrictions.

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
