# Peer review of "Design of a Compact Circularly Polarized Implantable Antenna for Capsule Endoscopy Systems"

_sensors, 2024, doi:10.3390/s24123960_

Round 1
Reviewer 1 Report
Comments and Suggestions for Authors
This work can be accepted after addressing the comments of this reviewer. Major revision is recommended.
In this work, the authors proposed a miniature circularly polarized implantable antenna designed for wireless capsule endoscopy systems operating at 2.45 GHz. This work is of interest, however, there are some concerns of this reviewer to be addressed. Please find below my comments:
· How did you determine the optimal dimensions for the chamfer rectangular slot and the curved radiation structure? A parametric analysis is required to get more insight about it.
· Can you elaborate on the simulation settings and how accurately they reflect the dielectric properties of real human tissues ? What were the real challenges during this work as most of the substrate might not available in open literature. This analysis will really help the researchers working in this domain.
· The effective AR bandwidth is reported as 120 MHz (2.38 to 2.50 GHz). How this bandwidth was achieved. What are the factors that help in increasing the AR bandwidth of the proposed antenna desing.
· What are the key challenges in the manufacturing process of this antenna, and how do you ensure consistency and reliability in mass production? What are its impacts on real human body and how the performance might degraded or effected.
· What are the next steps for improving the antenna design, particularly in terms of enhancing bandwidth, reducing size further, and improving robustness in more varied and dynamic environments? This can lead to future work directions of researchers working in this domain.
Comments on the Quality of English Language
Moderate English language edits are required.
Author Response
Thank you for your hard work. Your suggestions are of great help in improving the quality of the paper. We have made individual modifications to your suggestions, please refer to the attached document for details.

Reviewer 2 Report
Comments and Suggestions for Authors
The authors present the design of a CP antenna for endoscopic capsules, which is a very challenging work due to the difficult and constantly-changing environment of operation.
I think the work is good but the presented results and explanations are not sufficient. There are some deficiencies that must be addressed prior to publication. After this, I'd be glad to revisit this work.
My comments are given below, ordered by the manuscript line.
78: Please specify the used simulator and solver.
141: Please elaborate on the dimensions chosen for the biological tissue. Were they obtained in the bibliography? If so, add references to support.
154: In fig2, it is not clear what the results portray. It looks like the antenna performance was evaluated in each tissue individually, but the text does not explain this. Please clarify.
154: The axial ratio values were obtained in what direction of the antenna? Please provide a figure with the antenna and a coordinate system for reference. It is also useful for fig9 for example.
206: In fig4, the red and pink lines have the same subtitles.
206: In fig4,5,6,7 what is the antenna's surrounding environment? Please clarify in the document.
263: Specify the limit value according to the norm.
266: Please clarify the sentence "Hence, the maximum allowable value is much lower than the transmit power of commercial transm". Do you mean that the commercial transmitters use power levels below the maximum allowed power for your antenna?
269: Please add to fig 9 the model of the simulation so that we can understand the directions and the context.
280: What are the dimensions of the minced pork bowl? Where in the bowl is the antenna placed?
The simulation environment needs to be updated to match the measurement. It is not useful to compare a simulation and a measurement in completely different environments, as the results will inevitably disagree.
290: Can you obtain a measured radiation pattern in one plane where the bowl can rotate? It would greatly value the manuscript.
295: The conclusion needs improvement, particularly the description of the proposed antenna.
Author Response

(The authors gave the same response as above.)

Round 2
Reviewer 1 Report
Comments and Suggestions for Authors
Thank you for addressing all the concerns of this reviewer. I have no more comments.
Author Response
We appreciate your hard work, and your constructive suggestions are very helpful in improving the quality of the paper. Thank you very much.
Reviewer 2 Report
Comments and Suggestions for Authors
Thank you for the revision. I just have a few details to iron out.
Q4: Please add a figure with the antenna oriented in the carthesian referencial so that we know where theta=0 is relatively to the antenna geometry.
Fig 9: The subtitle is incorrect, it is simulated, not measured.
Q8: missing the model of the antenna with the coordinate system for clarification. Can link to my Q4 if the referencial does not change.
Q9: Please add this clarification to the paper.
Q10: If so, it should still be possible to measure the gain of the antenna in the E-plane, as long as you rotate the TX antenna centered around the implantable antenna. This would greatly value the manuscript.
Author Response

(The authors gave the same response as above.)
